# Structural and Molecular Basis for Mitochondrial DNA Replication and Transcription in Health and Antiviral Drug Toxicity

**DOI:** 10.3390/molecules28041796

**Published:** 2023-02-14

**Authors:** Joon Park, Noe Baruch-Torres, Y. Whitney Yin

**Affiliations:** 1Department of Biochemistry and Molecular Biology, Sealy Center for Structural Biology and Molecular Biophysics, University of Texas Medical Branch, Galveston, TX 77555, USA; 2Department of Pharmacology and Toxicology, Sealy Center for Structural Biology and Molecular Biophysics, University of Texas Medical Branch, Galveston, TX 77555, USA

**Keywords:** mitochondria, DNA replication, RNA transcription, human diseases, antivirals, HIV, HCV

## Abstract

Human mitochondrial DNA (mtDNA) is a 16.9 kbp double-stranded, circular DNA, encoding subunits of the oxidative phosphorylation electron transfer chain and essential RNAs for mitochondrial protein translation. The minimal human mtDNA replisome is composed of the DNA helicase Twinkle, DNA polymerase γ, and mitochondrial single-stranded DNA-binding protein. While the mitochondrial RNA transcription is carried out by mitochondrial RNA polymerase, mitochondrial transcription factors TFAM and TFB2M, and a transcription elongation factor, TEFM, both RNA transcriptions, and DNA replication machineries are intertwined and control mtDNA copy numbers, cellular energy supplies, and cellular metabolism. In this review, we discuss the mechanisms governing these main pathways and the mtDNA diseases that arise from mutations in transcription and replication machineries from a structural point of view. We also address the adverse effect of antiviral drugs mediated by mitochondrial DNA and RNA polymerases as well as possible structural approaches to develop nucleoside reverse transcriptase inhibitor and ribonucleosides analogs with reduced toxicity.

## 1. Unique Replication Mechanism for Mitochondrial DNA

Human mitochondrial DNA (mtDNA) is approximately 16.6 kbp double-stranded, circular DNA that codes 37 genes including a subset of components in the oxidative phosphorylation electron transfer chain (ECT), tRNAs, and rRNAs that are essential for mitochondrial protein translation [1]. The entire ECT is assembled with components coded into the mitochondria and nucleus to couple oxygen consumption with ATP synthesis. Enzymes that contact mtDNA replication, transcription, and mitochondrial ribosome proteins are nuclear coded, thus coordination of gene expressions in the two organelles are critical for cellular energy supply, cell cycle control, and metabolism [2]. Maintenance of mtDNA has profound implications for human health and chronological aging [3,4,5,6,7,8,9,10].

Except for the 1 kbp control region, both strands of human mtDNA are gene-coding without introns. The two strands are named H- and L-strand owing to their unequal contents of purines and pyrimidines. The mitochondrial control region (CR) is a non-coding DNA region with high variability. The CR contains two promoters, HSP and LSP, for H-strand and L-strand transcription, respectively, and an origin for H-strand replication, O_H_. The origin for L-strand replication is located about 8000 bp from O_H_ (Figure 1) [11,12]. MtDNA replication is primarily carried out by a strand-displacement mechanism, which differs from the canonical leading and lagging strand-coupled synthesis in nucleus where two strands are synthesized in synchrony. In the strand-displacement replication model, the H-strand is synthesized solely from the O_H_, generating a long, single stranded DNA D-loop. The L-strand synthesis begins only after the H-strand synthesis reaches the O_L_ when two-third of the H-strand has been synthesized (Figure 2) [13,14]. In recent years, another mtDNA synthesis model called RITOLS has been suggested that may function under special circumstances [15].

In vitro, mtDNA replication can be reconstituted with a minimum replisome consisting of the Twinkle DNA helicase, DNA polymerase gamma (Pol γ), and mitochondrial single-stranded DNA-binding protein (hmtSSB). Despite mitochondria’s bacterial origin, enzymes involved in mtDNA replication and RNA transcription are of high similarity to that of T7 bacteriophages.

Twinkle helicase belongs to the helicase superfamily 4 (SF4) that includes well-studied T7 bacteriophage helicase (gp4) and *E. coli* DnaB [17,18]. SF4 helicases unwind double-stranded DNA in a 5′-3′ direction, powered by nucleotides ATP, UTP, or dTTP hydrolysis [19].

Structurally, T7 gp4 adopts hexameric and heptameric ring configuration where the hexamer is thought to be the active form and the heptamer as the storage form [20,21,22]. Each subunit can be divided into the N-terminal domain (NTD) and C-terminal domain (CTD) connected by a linker [22,23]; the NTD contains primase activity, and CTD contains helicase activity (Figure 3a).

Sequence analysis readily enables identification of high homology between mammalian Twinkle’s CTD and T7 gp4 helicase domain, whereas the NTD of the two helicases are of less similarity. A low-resolution cryo-EM structure of a wild-type Twinkle [24,25] as well as a high-resolution disease mutant (W315L) [25] have been determined. The structures show that the Twinkle CTD adopts a nearly identical structure as that of the gp4 CTD, which is the main driving force for the ring configuration (Figure 3c).

Despite the NTDs of the two helicases being less conserved in sequence, they still share structural similarity (Figure 3d). The T7 gp4 NTD contains primase activity, but Twinkle is void of enzymatic activity. Structural comparison illustrated that the catalytic residues for primase in gp4 are substituted with amino acids, explaining its lack of primase activity (Figure 3d).

However, the two helicases differ in domain organization: in T7 gp4, the NTD and CTD form head-to-tail interaction, whereas forming side-by-side configuration in Twinkle (Figure 3a,b). Thus, the ring thickness of gp4 (~100 Å) is greater than that of Twinkle (~60 Å). While T7 replisome shed many important lights on strand-coupled DNA replication, it may not represent that in mitochondria. First, the domain arrangement suggests the replisome structure of human mitochondria replisome might be different from that of the T7; second, the T7 replisome consists of one gp4 helicase and two DNA polymerases (T7 DNAP), where the gp4 NTD interacts with the lagging strand T7 DNAP and CTD with the leading strand T7 DNAP. As mitochondrial dsDNA is replicated asymmetrically, mitochondrial replisome could function sufficiently with one Twinkle helicase and one Pol γ, and the NTD of Twinkle interacts with mtSSB.

Pol γ is the only DNA replicase in mitochondria. The holoenzyme consists of a catalytic subunit Pol γA that contains active sites for 5′-3′ polymerization (*pol*) for DNA synthesis and 3′-5′ exonuclease (*exo*) for proofreading to correct erroneously incorporated nucleotides, and an accessory subunit Pol γB that is void of intrinsic enzymatic activity but can regulate all activities of Pol γA. Pol γA belongs to the high-fidelity DNA Polymerase A family, whose members include T7 DNA polymerase and *E. coli* Pol I. Like many members of this polymerase family, Pol γA alone exhibits low processivity, i.e., lack of ability to replicate long genome efficiently; the feature renders these polymerases unsuitable to function as DNA replicases. Pol γA contains an additional large spacer domain sandwiched between *pol* and *exo* domains that constitutes a major binding site for the accessory subunit Pol γB. Upon association with Pol γB, the processivity and catalytic rate of the Pol γ holoenzyme is significantly increased [26,27]. Deletion of the spacer domain in drosophila Pol γA drastically reduced subunit interaction and the processivity of the holoenzyme [28]. Pol γ also contains a 5′-deoxyribose phosphate lyase activity which contributes to the base excision repair, an important pathway in mtDNA repair and maintenance by reducing mutation rates [29].

Accessory subunit Pol γB structurally resembles the class II tRNA synthetases, specifically the threonyl tRNA synthetase [30,31,32]. The accessory subunit is a dimer in mammals, a monomer in insects, and outright missing in single cell eukaryotes [31,33,34,35,36,37]. Nevertheless, the catalytic residues of synthetase are not conserved in Pol γB, explaining the lack of catalytic activity. Each subunit of Pol γB performs separate functions: the proximal monomer enhances DNA binding, while the distal monomer enhances processivity [38]. Interestingly, the origin O_L_ is surrounded by tRNA genes, and it has been hypothesized that the tRNA could serve as primers for replication of the L-strand.

Human mtSSB (hmtSSB) is a tetrameric protein that binds to the single-stranded DNA generated by asymmetrical DNA synthesis from O_H_, reducing secondary structures of the nascent DNA strand in serving for replication. However, hmtSSB appears to function beyond passive DNA interaction, as it exerts synergy in the replisome with Pol γ and Twinkle helicase [39,40]. Biochemical studies have shown that mtSSB stimulates both polymerase and exonuclease activities in Pol γ via either protein-protein interaction or organization of DNA topology [41,42,43]. In addition, mtSSB has been shown to play an important role in optimizing RNA primer for DNA replication on both O_H_ and O_L_ [44].

Individually, human Pol γ holoenzyme and Twinkle do not have the desired activities for replicating dsDNA: Pol γ lacks strand-displacement synthesis and can only replicate on single-stranded DNA template, while Twinkle exhibits minimal DNA unwinding ability. In fact, human Twinkle acts as a DNA annealer and its apparent ‘unwinding’ is due to its DNA strand switching [45,46]. Nonetheless, the two proteins together are capable of replicating dsDNA synthesis, suggesting the helicase and polymerase’s strand-displacement synthesis are mutually stimulative. The addition of hmtSSB significantly enhances polymerase/helicase DNA synthesis at the replication fork [39,40]. When replicating a duplex DNA template, the addition of mtSSB significantly increases the combined activity of the polymerase and helicase, despite the lack of direct interaction between the proteins. This behavior is similar to the replication systems in bacteria and bacteriophage.

Leading-strand mitochondrial DNA replication is done primarily by Pol γ, Twinkle, and mtSSB. Under the strand-displacement model, Pol γ and Twinkle displaces the parental strand and synthesizes the daughter strand, while mtSSB binds to the displaced strand.

## 2. Replication Initiation-Coupling of RNA Transcription with DNA Replication

In nucleus, initiation of DNA replication system begins with RNA from RNA primer synthesis by a designated primase. However, primase has not been found in mammalian mitochondria, thus making the short RNA primer transcribed by the mitochondrial RNA polymerase (POLRMT) serve as a primer for DNA replication, similar to that of the bacteriophage systems. Although the exact switch mechanism between transcription and replication priming is unknown, it has been suggested that the exonuclease activity in POLRMT is required for replication priming in a fruit fly [47]. Pol γ begins DNA replication on this RNA primer, creating an RNA-DNA hybrid which has been found in both human and mouse cells [48]. This RNA portion of the hybrid is then efficiently processed by RNase H1 and EXOG [49,50] as well as by FEN1, DNA2, and MGME1 to a lesser extent [51].

Both mtDNA strands code for genes and are transcribed from HSP and LSP, respectively. Although transcription from the two promoters are symmetrical and transcribed at the same rate [52], transcripts of the two strands have a different half-life: transcripts of the HSP have a faster turnover and no accumulation relative to the transcripts from LSP [53,54,55,56]. The L-strand transcription promoter (LSP) is located upstream to the O_H_ in the D-loop. Persistent RNA/DNA hybrids formed with the L-strand corresponded almost exclusively to the right half of the genome past the O_H_. By contrast, the hybrid involving the H-strand appeared to be localized in half of the genome, particularly in the region adjacent of O_H_ [57]. Transcripts are usually polycistronic and near the length of the genome and are subsequently processed, except that at three Consensus Sequence Blocks, CSBI, CSBII, and CSB III. The POLRMT has a high probability of terminating transcription at the CSBII, located between the LSP promoter and O_H_.

Mitochondrial RNA transcription system possesses hybrid features of prokaryotic and eukaryotic systems. Mitochondrial RNA polymerases contain clear features of bacteriophages, specifically T7 and T3 [19,58,59]. However, unlike its single subunit bacteriophage counterparts that sufficiently catalyze RNA transcription reaction, including promoter recognition and unwinding, binding to nucleotides, formation of phosphodiester bond, and transition from initiation to elongation transcription, POLRMT alone is unable to recognize its promoters, thereby depending on its transcription factors, TFAM and TFMB2, for promoter recognition and transition from initiation to elongation RNA synthesis.

Mitochondrial transcription can be reconstituted in vitro with POLRMT, transcription factor A (TFAM), transcription factor B2 (TFB2M), and transcription elongation factor (TEFM). POLRMT shares high sequence homology with T7 RNA polymerase. Unlike the self-sustained T7 RNAP, POLRMT needs transcription factors for promoter recognition. TFAM binds to the mitochondrial promoter’s sequence and creates a stable protein-DNA complex but can also act as bender or DNA unwinder in non-specific DNA sequences [60]. TFAM and TFB2M works synergistically to support mitochondrial transcription initiation [61]. TEFM has been shown to increase processivity of the transcription and to stop premature transcription termination at CSB II [62,63]. Structures of the mitochondrial transcription initiation and elongation complexes revealed the mechanism in which TFAM recruits POLRMT, TFB2M opens the DNA duplex and stabilizes POLRMT, and TEFM forms a sliding clamp to increase the processivity of POLRMT [64,65].

Previous investigations suggested that CSBII improves the stability of RNA-DNA hybrid. This agrees with other experimental findings in which these hybrids are found close to the CSB II region where the transition from RNA primers to DNA replication occurs [66]. Over 95% of the H-strand transcription initiation synthesis are prematurely terminated around 600 bp downstream of the initiation point [67]. Another study reports a shorter, prematurely terminated product about 100 bp downstream of the LSP promoter that coincides with the CSBI, CSB II, and CSB III region [68]. By using a mutational approach, CSB II was discovered to play an important role in premature transcription termination with little to no role from CSB I and CSB III [68].

The terminated RNA transcripts at CSBII can serve as primers for replication at O_H_ so that the Pol γ replaces the RNA polymerase and begins DNA synthesis from the RNA primers; perhaps, DNA polymerase has higher affinity to the transcription bubble than the RNA polymerase once POLRMT dissociates from the template. Thus, mitochondrial RNA transcription and DNA replication are intertwined; both RNA transcription and DNA replication machineries control mtDNA copy numbers, cellular energy supplies, and cellular metabolism.

## 3. Mutations of Pol γ and Twinkle Implicated in Multi-System Human Diseases

Maintenance of mitochondrial DNA (mtDNA) homeostasis is critical in maintaining cellular energy supplies and cell cycle controls. Mutations in nuclear genes involved in mtDNA homeostasis, i.e., *POLG* and *POLG2* (code for Pol γA and PolγB, respectively), *TWNK* (codes for Twinkle helicase), and *TYMP* (codes for thymidine phosphorylase) result in mtDNA depletion or deletions in post-mitotic tissues, leading to clinical presentations collectively termed “mtDNA maintenance disorders.” The symptoms of mtDNA maintenance disorders predominantly manifest as muscle weakness, central nerve system involvement, and hepatic dysfunction.

### 3.1. Pol γ Replication Fidelity and Aging

Mitochondrial DNA integrity is directly implicated in the aging process. Mutation in Pol γ exonuclease active site would abolish the proofreading ability and increase replication error [69,70]. Transgenic mice carrying an exonuclease-deficient (exo^−^) Pol γ variant exhibited increased mtDNA mutations and premature aging syndromes [71]. Similar observations were made in *Drosophila* carrying Pol γ exo^−^ allele, but when polymerase-deficient (pol^−^) Pol γ allele was expressed in trans with Pol γ exo^−^, it reversed the aging characteristics [72], establishing a direct connection between Pol γ proofreading deficiency and aging. Pol γ exo^−^ variant has not been found in nature; nonetheless, oxidative stress can selectively hamper wild-type Pol γ exonuclease activity, generating a phenotype similar to the exo-deficient polymerase [73]. The study established a correlation between organelle oxidative state and aging.

### 3.2. Pol γ Mutations Implicated in Diseases

Over 150 mutations in Pol γA have been reported from patients such as Alpers-Huttenlocher Syndrome (AHS) [74,75,76], progressive external ophthalmoplegia (PEO) [77], and childhood myocerebrohepatopathy spectrum (MCHS) [78]; the ataxia neuropathy spectrum (ANS) [79]; sensory ataxic neuropathy and dysarthria (SANDO) [80] (Figure 3a). A growing number of patients with *POLG* mutations resulted in multiple mtDNA deletions respiratory chain deficiency, leading to co-segregation of PEO, parkinsonism [81].

The disease-causing Pol γ mutations are better understood using high resolution 3D structures [82,83]. Their impact on the polymerase activity can be illustrated by the mutant locations in functional domain. For instance, a mutation in the *pol* site of Pol γA directly affect Pol γ DNA synthesis catalysis [84,85,86,87] (Figure 4a,b), and a mutation in the thumb domain reduces DNA binding affinity [88] (Figure 4a,c). Another class of mutations are located interface of the PolγA and PolγB (Figure 4d). The mutant holoenzyme showed reduced subunit interaction, causing lower synthesis processivity and decreased synthesis. Biochemical characterization of subunit interface mutation showed significantly reduced polymerase’s DNA synthesis [89].

### 3.3. Twinkle Disease Mutations

To date, over 40 Twinkle mutations have been associated with human clinical disorders (Figure 5a), such as progressive external ophthalmoplegia and ataxia neuropathies among other mitochondrial diseases. Disease variants of Twinkle helicase can lead to increased mtDNA copy number [59], mtDNA depletion, and accumulation of replication intermediates [90].

Structural analyses provide in-depth understanding of the defects of Twinkle mutants. The oligomeric structure of wild-type Twinkle is regulated by salt concentration, Mg^2+^. and ATP; the latter strengthens Twinkle’s hexameric ring configuration [45]. Using T7 gp4 hexameric structure from a replisome and AlphaFold2 [91], a hexameric human Twinkle was constructed by superposition of the C-terminal helicase domain of the two proteins, which provide a structural basis for analyses of the disease mutations.

One class of mutants is in the subunit interface. These include in-frame duplication of amino acids 353–365, which caused large mtDNA deletion and familial parkinsonism and ophthalmoplegia (PEO) [92], R374Q [93]; and mutations in the linker region (A359T, I367T, S369P, R374Q and L381P), which reduce ATP hydrolysis and abolish DNA helicase activity [94,95]. These disease mutants are located at the subunit interface (Figure 5b), and their substitutions could disrupt subunit interface and alter the oligomeric states of the helicase. Some disease-associated mutants that are not located on the subunit interface in the predicted structure still altered oligomeric states, e.g., W318 induces formation of heptamer and octamer [25,94]. The above mutants displayed altered Twinkle oligomeric state, to distorted heptamer (R334Q) to mixture of hexamers, heptamers, octamers, and monomers (P335L) [10,94]. The mutations imply that the maintenance of proper hexametric structure is critical to its correct function.

Another class of mutants are in the Twinkle NTD and CTD interface. Twinkle domain interface is more extensive than that in T7 gp4. NTD mutants (R265C, A318T, R334Q, and P335L) and CTD mutations (V507I and Y508C) are located at the domain interface, and their interactions are potentially important in stabilizing the domain structures (Figure 5c). The mutants manifested dysfunction, implying the functional necessity for domain interactions.

## 4. Pol γ and POLRMT as Drug Adverse Reaction Targets

Perhaps due to the homology to viral counterparts of mitochondrial DNA replication and transcription machineries, among all human polymerases, Pol γ and POLRMT are selectively inhibited by antiviral drugs. Pol γ cross-reacts with nucleoside analog drugs designed against human immunodeficiency virus reverse transcriptase (HIV RT) [96,97,98], whereas POLRMT cross-reacts with drugs against hepatitis C virus (HCV) RNA-dependent RNA polymerase [99]. The main characteristic of nucleoside reverse transcriptase inhibitors (NRTIs) is the lack of 3′-OH group, which prevents further nucleotidyl-transfer reactions and acts as chain terminator for HIV RT. However, the earlier generation of nucleoside analog inhibitors such as zalcitabine (ddC), didanosine (ddI), and stavudine (d4T) are particularly toxic to Pol γ because Pol γ is unable to discriminate against 2′,3′-dideoxynucleotides (ddNTPs) and incorporates ddNTPs with nearly equal efficiency as the natural substrate, dNTPs [100,101]. The later generations of inhibitors displayed much reduced toxicity because the human polymerase appear to possess more stringent nucleotide selectivity toward nucleoside modifications than the viral counterpart [102]. Compared to the HIV RT, Pol γ displays greater sensitivity and discrimination toward ribose modifications on NRTI. For example, 4′-ethynylstavudine (4′-Ed4T), which is a 4′-carbon-substituted form of stavudine, effectively inhibit HIV RT with similar efficiency as stavudine but with less cellular toxicity [103]. In pre-steady-state kinetic studies, stavudine was shown to be incorporated similarly to dTTP in both Pol γ and HIV RT, whereas 4′-Ed4T was highly discriminated (lower incorporation of nucleoside analog compared to regular nucleotide) by Pol γ while showing similar incorporation efficiency as stavudine on HIV RT [104]. Another example of ribose modification influencing nucleotide incorporation is zidovudine (AZT). Zidovudine has 3′-substitution with azido group, replacing the 3′-OH. HIV RT incorporates zidovudine only slightly less than dTTP, whereas Pol γ discriminated zidovudine by roughly 37,000-fold [105,106]. In addition, different nucleoside analogs show different excision rates by the exonuclease activity [106,107]. Mutations in human Pol γ relaxed its selectivity and increased its incorporation of nucleotide inhibitors [108]. A clinical study conducted with AIDS patients revealed that patients with elevated drug toxicity carried significantly higher Pol γ mutations, suggesting Pol γ genotype should be a critical factor in the treatment of HIV infection. Thus, understanding the mechanisms behind differential incorporation and excision of nucleoside analogs based on modifications and the effect of Pol γ mutations is important for developing NRTI with reduced toxicity.

The success in the development of a cure for HCV infections is based on ribonucleoside inhibitors. Unlike NRTI, ribonucleoside inhibitors contain 3′-OH and are non-obligate chain terminators, yet they terminate HCV RNA-dependent RNA polymerase by sterically hindering further elongation of RNA synthesis. Many ribonucleoside inhibitors exhibit cross inhibition of POLRMT; interestingly, these inhibitors do not interact with Pol γ. Considering abnormalities in mitochondria as prominent features of HCV clinical disease, where ultrastructural changes, electron transport are altered, and excess reactive oxygen species (ROS) are produced [109,110], reducing mitochondrial toxicity is fundamental to successful anti-HCV drug design.

Although Pol γ and POLRMT are adverse reaction targets to antivirals, they are also a good drug target for other diseases. For instance, inhibiting Pol γ with Congo red resulted in reduced mitochondrial base-excision repair, which selectively suppressed MutL homolog 1 (MLH1)-deficient tumor growth [111]. Additionally, toxicity toward antivirals has been taken advantage of to reduce tumor growth. Zalcitabine (ddC), one of the earlier NRTIs that was later withdrawn due to high mitochondrial toxicity, preferentially inhibited mtDNA replication in acute myeloid leukemia (AML) cells with 10-fold less concentration of zalcitabine required to inhibit mtDNA replication in HEK 293 cells [112]. Similarly, inhibiting POLRMT with 2′-C-methyladenosine (2-CM), a potent inhibitor of HCV RNA-dependent RNA polymerase that also targets POLRMT, reduced mitochondrial gene expression in conjunction with AML tumor growth in a concentration-dependent manner [113]. Similar trend was also observed in following cancer cells: chronic myelogenous leukemia, leukemic monocyte lymphoma, non-Hodgkin lymphoma, and promyelocytic leukemia [113]. Inhibiting mtDNA maintenance at replication, repair, or transcription level ultimately impairs oxidative phosphorylation, which is detrimental to cancer cells.

Structural work on human Pol γ and POLRMT in complex with corresponding nucleoside inhibitors are scarce but necessary to complement the biochemical and kinetic data available on antivirals to ultimately design nucleoside inhibitors with little-to-no toxicity. This approach will also allow better design of antivirals to target drug-resistant mutants that are still discriminated by human polymerases as we gain better knowledge of the discrimination mechanism in each enzyme.

## 5. Concluding Remarks and Futures Perspectives

Mitochondrial genome integrity is essential for cellular energy supply, cell cycle, and metabolism. Aberrant human mitochondrial replication has been associated with neurological, muscular, and cardiovascular diseases, as well as aging. Mitochondrial gene replication and transcription machineries possess combined features of that from prokaryotes and eukaryotes. The unique combination diverges the mitochondrial system from its nuclear counterpart. Future investigations are needed to reveal the unique mechanisms governing human mitochondrial DNA replication, transcription, and maintenance. Specifically, the synergy between Pol γ, Twinkle helicase, and single-stranded DNA binding protein in DNA replication and the signals that switch DNA Pol γ between replication and repair, and POLRMT between mRNA gene transcription and short RNA primers synthesis. The functional condensation is reminiscent of that in prokaryotes. Perhaps for these reasons, nucleoside/nucleotide analog antiviral drugs designed against viral polymerases are specifically inhibitory to mitochondrial polymerases; the resulted drug toxicity imposes a major hurdle for development of safe antiviral reagents.

Recent viral epidemics and pandemics imposed severe threats to public health. Effective and low-toxic antiviral drugs are in urgent need. Nucleoside/nucleotide analog inhibitors are among the most effective antiviral reagents. Nevertheless, their efficacy must be balanced with low toxicity to human mitochondrial polymerases. Understanding and eventually eliminate drug toxicity are critical steps of drug design. Future structural and functional studies of antiviral interactions with a viral polymerase should be combined with the host enzymes to guide development of effective and low-toxic drugs.

## Figures and Tables

**Figure 1 molecules-28-01796-f001:**
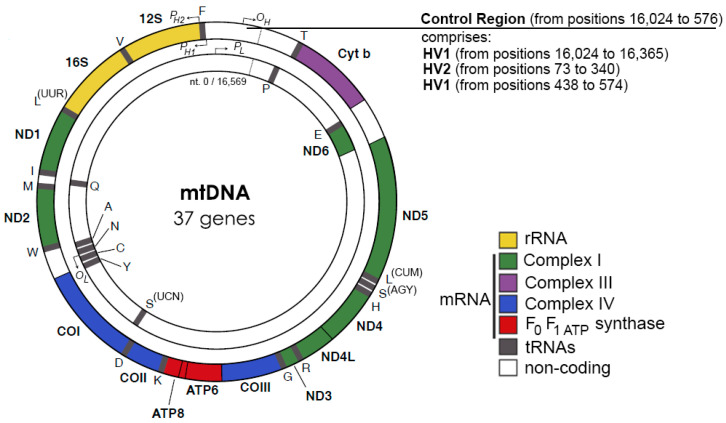
Human mtDNA genes, adopted from [11,12].

**Figure 2 molecules-28-01796-f002:**
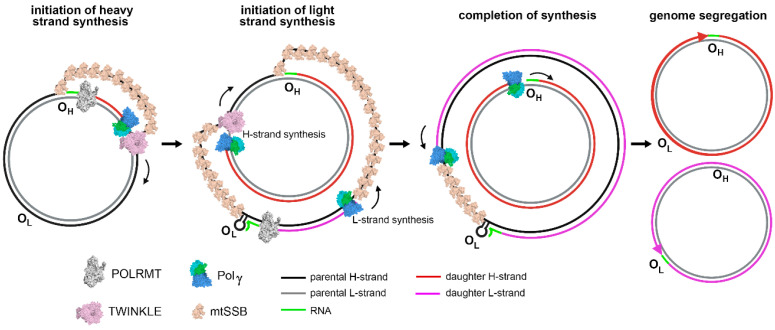
mtDNA replication: initiation and completion of synthesis in strand-displacement fashion, adopted from [16].

**Figure 3 molecules-28-01796-f003:**
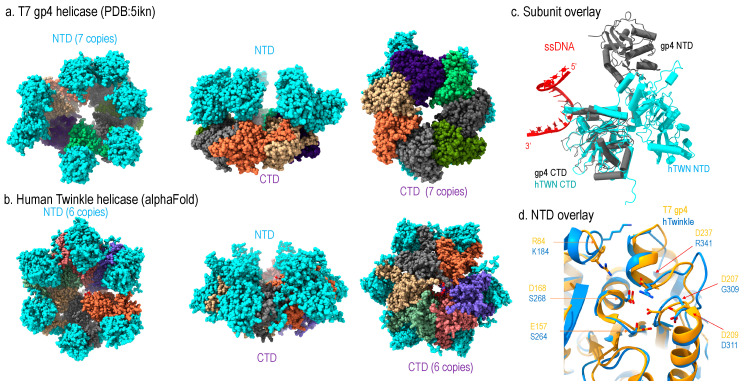
Structural comparison of human Twinkle helicase with T7 gp4 helicase (**a**) structure of apo heptameric gp4, the NTD is colored in cyan and CTD in multi-color, (**b**) Alphafold-predicted human helicase structure in similar color theme, (**c**) subunit structure of Twinkle (cyan) overlayed with T7 gp4 (grey), (**d**) Comparison of NTD structures of Twinkle (blue) and T7 gp4 (yellow).

**Figure 4 molecules-28-01796-f004:**
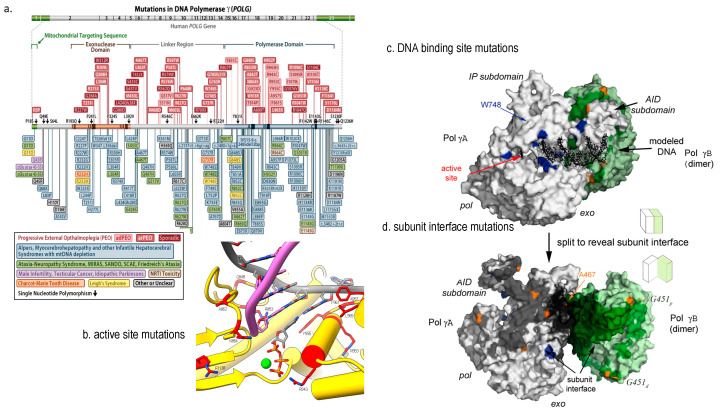
Structural view of Pol γ mutations. (**a**) disease-associated mutations depicted on a linear sequence of Pol γA (adopted from https://tools.niehs.nih.gov/polg/_assets/images/POLG_mutation.png (accessed on 10 December 2022), (**b**) mutations in Pol γA active site, (**c**) mutations in DNA binding sites, and (**d**) mutations in subunit interface, where Pol γA is colored white, proximal and distal Pol γB in green and light green, respectively, and DNA in black adopted from [82]. © 2009 Elsevier Inc. Published by Elsevier Inc.

**Figure 5 molecules-28-01796-f005:**
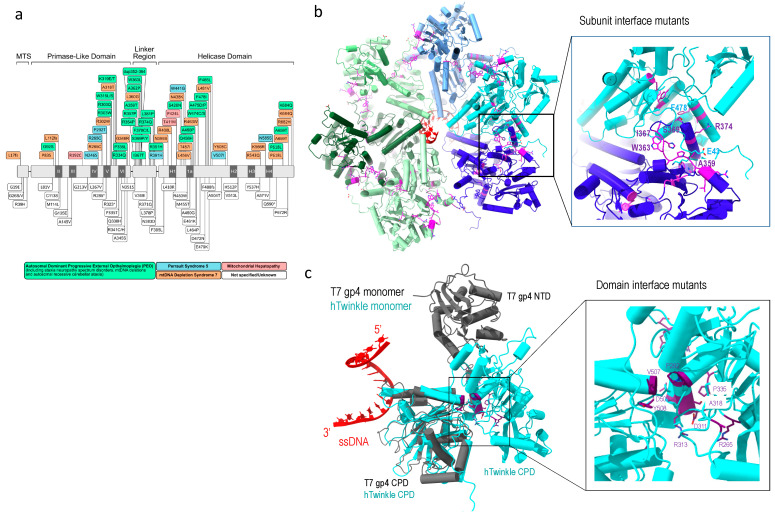
Twinkle disease-implicated mutations. (**a**) Linear depiction of Twinkle mutations implicated in human diseases, adopted from [10], (**b**) subunit interface mutations (magenta), and (**c**) domain interface mutations (purple), where Twinkle subunit is colored in cyan, T7 gp4 in grey, and single-stranded DNA in red.

## Data Availability

Not applicable.

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
