# Peer review of "Structural and Molecular Basis for Mitochondrial DNA Replication and Transcription in Health and Antiviral Drug Toxicity"

_molecules, 2023, doi:10.3390/molecules28041796_

Round 1

Reviewer 1 Report

Comments to author:

This manuscript provides short review of the structural biology of both mitochondrial DNA replication and transcription. Authors briefly discuss mitochondrial DNA replication and transcription machinery and then highlight structural features of mitochondrial DNA polymerase gamma and TWINKLE helicase. Authors describe mutations of Pol-gamma and TWINKLE implicated in human mitochondrial diseases. The review is well written and would be useful for the researchers working in this field. However, I have several commentaries that would require consideration.

Commentaries.

1. Authors might include DNA transcription in the title of the manuscript since they discuss its mechanisms in the text.

2. Authors should be aware when discussing mitochondrial DNA (mtDNA) repair. Human mitochondria lacks nucleotide excision and mismatch repair pathways. Despite elevated level of oxidative base damage mtDNA mutations lack the canonical ROS signatures (G->T/C->A & T->G/A->C) (Kennedy et al., 2013; Ju et al., 2014; see also doi.org/10.1080/24701394.2020.1734586). The studies on the DNA glycosylase deficient mice models Ogg1−/− knockout (Itsara et al. 2014) and on the Ogg1−/−Mutyh−/− double knockout (Halsne et al. 2012) failed to detect an increase in the spontaneous mutation rate in the mitochondrial genome, suggesting that base excision repair play minor role in mtDNA mutagenesis.

3. Page 6, Chapter “Pol gamma replication fidelity & aging”, lines 218-220, authors wrote: “Similar observations were made in  Drosophila carrying Pol gamma exo-, but knocking-in wild-type Pol gamma reversed the aging characteristics (Bratic et al., 2015), establishing a direct connection between Pol gamma proofreading deficiency and aging”.

The sentence written in misleading manner. It is not clear how knocking-in wild-type Pol gamma can reverse the aging characteristics? Authors forgot to mention that Bratic et al constructed Pol gamma (pol-) in addition to Pol gamma (exo-) mutant allele. When both allele are combined together, they could complement each other and avoid embryonic lethality. The sentence should be re-written and authors should mention Pol gamma (pol-) mutant allele construction.

4. Page 8, Chapter: Pol gamma and POLRMT as drug adverse reaction targets, lines 286-288, authors wrote: “Perhaps due to the prokaryotic origin of mitochondrial DNA replication and transcription machineries, among all human polymerases, Pol gamma and POLRMT are selectively inhibited by antiviral drugs”.

It is not clear how antiviral drugs can act on the proteins due to their bacterial origins? Antiviral drugs are efficient against viruses and viral proteins. It would be better to put “… due to the viral origin (or homology to viral proteins) of mito DNA replication and transcription, pol gamma and POLRMT are inhibited by antiviral drugs …”.

5. Page 6, line 212, change “hepatic disfunction” to “hepatic dysfunction”.

6. Authors do not provide abbreviation for “HCV”, I guess it is for the hepatitis C virus.

Author Response

Reviewer #1:
1. Authors might include DNA transcription in the title of the manuscript since they discuss its mechanisms in the text.

Included “transcription” in the title

  1. Authors should be aware when discussing mitochondrial DNA (mtDNA) repair. Human mitochondria lacks nucleotide excision and mismatch repair pathways. Despite elevated level of oxidative base damage mtDNA mutations lack the canonical ROS signatures (G->T/C->A & T->G/A->C) (Kennedy et al., 2013; Ju et al., 2014; see also doi.org/10.1080/24701394.2020.1734586). The studies on the DNA glycosylase deficient mice models Ogg1−/−knockout (Itsara et al. 2014) and on the Ogg1−/−Mutyh−/−double knockout (Halsne et al. 2012) failed to detect an increase in the spontaneous mutation rate in the mitochondrial genome, suggesting that base excision repair play minor role in mtDNA mutagenesis.

We thank the reviewer for bring this to our attention.

  1. Page 6, Chapter “Pol gamma replication fidelity & aging”, lines 218-220, authors wrote: “Similar observations were made in  Drosophila carrying Pol gamma exo-, but knocking-in wild-type Pol gamma reversed the aging characteristics (Bratic et al., 2015), establishing a direct connection between Pol gamma proofreading deficiency and aging”.

The sentence written in misleading manner. It is not clear how knocking-in wild-type Pol gamma can reverse the aging characteristics? Authors forgot to mention that Bratic et al constructed Pol gamma (pol-) in addition to Pol gamma (exo-) mutant allele. When both allele are combined together, they could complement each other and avoid embryonic lethality. The sentence should be re-written and authors should mention Pol gamma (pol-) mutant allele construction.

Corrected to “Similar observations were made in Drosophila carrying Pol g exo- allele, but when polymerase-deficient (pol-) Pol g allele was expressed in trans with Pol g exo-, it reversed the aging characteristics, establishing a direct connection between Pol g proofreading deficiency and aging.” P6 Lines 211-213

  1. Page 8, Chapter: Pol gamma and POLRMT as drug adverse reaction targets, lines 286-288, authors wrote: “Perhaps due to the prokaryotic origin of mitochondrial DNA replication and transcription machineries, among all human polymerases, Pol gamma and POLRMT are selectively inhibited by antiviral drugs”.

It is not clear how antiviral drugs can act on the proteins due to their bacterial origins? Antiviral drugs are efficient against viruses and viral proteins. It would be better to put “… due to the viral origin (or homology to viral proteins) of mito DNA replication and transcription, pol gamma and POLRMT are inhibited by antiviral drugs …”.

Rewritten to “Perhaps due to the homology to viral counterparts of mitochondrial DNA replication and transcription machineries,…” P8 Line 274.

  1. Page 6, line 212, change “hepatic disfunction” to “hepatic dysfunction”.

Corrected P6 Line213

  1. Authors do not provide abbreviation for “HCV”, I guess it is for the hepatitis C virus.

Corrected P8 Line 270

We also provided abbreviation human immunodeficiency virus for HIV P8 Line269.

Reviewer 2 Report

This review aims to summarize structural and molecular basis for mitochondrial DNA replication, the kind of topic has been seldom reviewed, and thus it is important for studying molecular biology. However, the preparation of the article was very poor. The writing of articles and image making need to be greatly strengthened, and discussion in this work are not well presented. Thus, I think this manuscript does not reach the standard of publication in molecules and needs to be reconsidered after major revision.

The subtitles in the manuscript are confusing, with only the first marked serial number. I think the following headings are not in parallel. The order of graphs is also chaotic in the manuscript. Two captions of Figure 2 occur, which is correct? Pol g holoenzyme is cited in the text but it is not ambiguous for this task. Legends in the Figure is inconsistent with the text, for example, OriH (in the text), an origin for H-strand replication, is OH (in Figure 1 and Figure 2)?

Abstract is not present in the submission file but is generally required.

The summary about structural and molecular basis for mitochondrial DNA replication is not thoroughly discussed. What are other enzymatic roles besides the pol r, twinkle helicase and mtSSB? There are some reported to-date that I have not found here.

Author Response

  1. The subtitles in the manuscript are confusing, with only the first marked serial number. I think the following headings are not in parallel. The order of graphs is also chaotic in the manuscript.

 We have numbered the rest of the subtitles in the text.

  1. Two captions of Figure 2 occur, which is correct? Pol g holoenzyme is cited in the text but it is not ambiguous for this task. Legends in the Figure is inconsistent with the text, for example, OriH (in the text), an origin for H-strand replication, is OH(in Figure 1 and Figure 2)?

We rewritten the legend as “mtDNA replication: initiation and completion of synthesis in strand displacement fashion”. We also fixed errors regarding figure 2.

We have unified OH and OL in the text and figure legends.

  1. Abstract is not present in the submission file but is generally required.

 We have included the Abstract.

  1. The summary about structural and molecular basis for mitochondrial DNA replication is not thoroughly discussed. What are other enzymatic roles besides the pol r, twinkle helicase and mtSSB? There are some reported to-date that I have not found here.

There are indeed many other proteins involved in mtDNA replication, it is not our intention to review the all replicating proteins, rather we focused on the ones implicated in human diseases and antiviral drug toxicity.

Round 2

Reviewer 2 Report

On the basis of the reports of the reviewers and my own consideration of the manuscript, the authors improved the manuscript, which can meet the requirements of publication in Molecules.